# Folic Acid Decreases Astrocyte Apoptosis by Preventing Oxidative Stress-Induced Telomere Attrition

**DOI:** 10.3390/ijms21010062

**Published:** 2019-12-20

**Authors:** Wen Li, Yue Ma, Zhenshu Li, Xin Lv, Xinyan Wang, Dezheng Zhou, Suhui Luo, John X. Wilson, Guowei Huang

**Affiliations:** 1Department of Nutrition and Food Science, School of Public Health, Tianjin Medical University, Tianjin 300070, China; liwen828@163.com (W.L.); mayue931224@163.com (Y.M.); 18222727318@163.com (Z.L.); lvxin@tmu.edu.cn (X.L.); wangxinyan@tmu.edu.com (X.W.); dezhengzhou@163.com (D.Z.); luosuhui@tmu.edu.cn (S.L.); 2Tianjin Key Laboratory of Environment, Nutrition and Public Health, Tianjin 300070, China; 3Center for International Collaborative Research on Environment, Nutrition and Public Health, Tianjin 300070, China; 4Department of Exercise and Nutrition Sciences, School of Public Health and Health Professions, University at Buffalo, Buffalo, NY 14214-8028, USA; jxwilson@buffalo.edu

**Keywords:** folic acid, astrocytes, apoptosis, telomere attrition, oxidative stress, in vitro

## Abstract

Astrocytes are the most widely distributed cells in the brain, and astrocyte apoptosis may play an important role in the pathogenesis of neurodegenerative diseases. Folate is required for the normal development of the nervous system, but its effect on astrocyte apoptosis is unclear. In this study, we hypothesized that folic acid (the therapeutic form of folate) decreases astrocyte apoptosis by preventing oxidative stress-induced telomere attrition. Primary cultures of astrocytes were incubated for 12 days with various concentrations of folic acid (0–40 μmol/L), then cell proliferation, apoptosis, intracellular folate concentration, intracellular homocysteine (Hcy) concentration, intracellular reactive oxygen species (ROS) levels, telomeric DNA oxidative damage, and telomere length were determined. The results showed that folic acid deficiency decreased intracellular folate, cell proliferation, and telomere length, whereas it increased Hcy concentration, ROS levels, telomeric DNA oxidative damage, and apoptosis. In contrast, folic acid dose-dependently increased intracellular folate, cell proliferation, and telomere length but it decreased Hcy concentration, ROS levels, telomeric DNA oxidative damage, and apoptosis. In conclusion, folic acid inhibited apoptosis in astrocytes. The underlying mechanism for this protective effect may be that folic acid decreased oxidative stress and thereby prevented telomeric DNA oxidative damage and telomere attrition.

## 1. Introduction

Telomeres are ribonucleoprotein structures capping the end of every linear chromosome [1]. They are composed of TTAGGG repeats coated with specific protecting proteins [2]. Telomeres maintain chromosome stability and integrity, regulate cellular proliferation, and prevent chromosome end fusion [3]. The shortening of telomeres with each cell division results eventually in chromosomal instability and activation of DNA damage response pathways that induce cell dysfunction and apoptosis [4]. Heritable factors (i.e., genetics) and environmental factors (i.e., malnutrition) influence telomere attrition [5,6]. Notably, oxidative stress accelerates telomere attrition [7,8,9,10].

Folate (vitamin B9) acts as a coenzyme to transfer one-carbon units that are necessary for deoxythymidylate synthesis, purine synthesis, and various methylation reactions. For example, folate is an important co-factor in the re-methylation process of homocysteine (Hcy) metabolism [11]. Consequently, folate deficiency leads to Hcy accumulation. Excessive Hcy, in turn, may cause oxidative stress in neurodegenerative disorders [12,13,14]. For instance, high circulating levels of Hcy (i.e., hyperhomocysteinemia) is a risk factor for Alzheimer’s disease [15,16]. 

Folic acid is the form of folate that is used therapeutically. It stimulates the proliferation of neural stem cells and the differentiation of neurons in vitro and in vivo [17,18]. Folic acid also supports the survival and proliferation of Schwann cells and PC-12 cells and it induces the migration of Schwann cells and astrocytes [18]. Folate metabolism is associated with the survival of astrocytes in the brain, as evidenced by the observation that methotrexate induces astrocyte apoptosis by disrupting folate metabolism [19]. However, whether folic acid can protect against astrocyte apoptosis is unclear. This knowledge gap is important because astrocytes are the most widely distributed cells in the brain and astrocyte apoptosis may contribute to the pathogenesis of neurodegenerative disorders such as Alzheimer’s disease [20,21]. The present study tested the hypothesis that folic acid decreases oxidative stress-induced astrocyte apoptosis by preventing telomere attrition. 

## 2. Results

### 2.1. Folic Acid Increased Cell Proliferation and Decreased Apoptosis

Incubation for 12 days with various concentration of folic acid (0–40 μmol/L) had multiple effects on astrocytes. Folic acid deficiency decreased cell proliferation compared with high-dose folic acid treatments (20 or 40 μmol/L) (Figure 1a). Folic acid increased cell proliferation in a dose-dependent manner (*r* = 0.756, *p* = 0.004). Cell proliferation rate in the high-dose folic acid group (40 μmol/L) was 1.1 times that in the normal folic acid group (10 μmol/L, *p* < 0.05, Figure 1b). Cell apoptosis was measured by flow cytometry and fluorescence staining with Hoechst 33342. Flow cytometry data revealed that folic acid dose-dependently decreased astrocyte apoptosis (*r* = −0.799, *p* = 0.002), since the total percent apoptosis rates in the presence of folic acid at 0, 10, 20, or 40 μmol/L were, respectively, 14.33 ± 4.2, 6.42 ± 3.00, 5.15 ± 2.79, 3.78 ± 3.10 (Figure 2a,c). Hoechst 33342 staining revealed a similar trend (*r* = −0.928, *p* = 0.000), since the total percent apoptosis rates in the presence of folic acid at 0, 10, 20 or 40 μmol/L were, respectively, 16.73 ± 2.03, 7.64 ± 0.53, 6.57 ± 0.76, 5.06 ±0.91 (Figure 2b,d). Taken together, these results indicated that folic acid decreased apoptosis and increased cell proliferation in a dose-dependent manner.

### 2.2. Folic Acid Decreased Intracellular Hcy and ROS

After 12 days of intervention, the intracellular folate concentration was very low in the folic acid-deficient group, while a high dose of folic acid (40 μmol/L) increased intracellular folate compared with the normal dose of folic acid (10 μmol/L) (*p* < 0.05) (Figure 3a). The intracellular Hcy concentration increased in the folic acid-deficient group, but a high dose of folic acid (40 μmol/L) decreased intracellular Hcy compared with the normal dose (10 μmol/L) (Figure 3b). Similarly, the intracellular reactive oxygen species (ROS) level was raised by folic acid deficiency and lowered by a high dose of folic acid (40 μmol/L) (Figure 3c).

### 2.3. Folic Acid Inhibited Telomeric DNA Oxidative Damage

The incorporation ratio of 8-oxoG within telomeric DNA was increased by folic acid deficiency and decreased by high folic acid concentrations (20 and 40 μM) in a dose-dependent manner (*r* = −0.907, *p* = 0.000) (Figure 4). The ratio of telomere dysfunction-induced foci (TIF)-positive cells in the folic acid deficiency group was 2.83 times that in the normal folic acid group (10 μmol/L, *p* < 0.05, Figure 5). High concentrations of folic acid decreased the abundance of TIF-positive cells dose-dependently (*r* = −0.950, *p* = 0.000) (Figure 5).

### 2.4. Folic Acid Inhibited Telomere Shortening

Southern blot analysis of telomere length revealed shortening by folic acid deficiency compared with treatments with normal and high folic acid concentrations (Figure 6a,b). However, there was no significant difference in telomere length between the normal-dose folic acid group (10 μmol/L 19.26 ± 2.85 kb), the medium-dose folic acid group (20 μmol/L, 20.98 ± 1.35 kb), and the high-dose folic acid group (40 μmol/L, 22.37 ± 2.19 kb). RT-PCR analysis confirmed that folic acid deficiency decreased telomere length but also showed that high folic acid concentrations increased telomere length dose-dependently (*r* = 0.885, *p* = 0.000) (Figure 6c).

## 3. Discussion

The present study determined that folic acid deficiency in primary astrocytes decreased intracellular folate, telomere length, and cell proliferation but increased Hcy concentration, ROS levels, telomeric DNA oxidative damage, and apoptosis. In contrast, incubation with folic acid for 12 days dose-dependently increased cell proliferation and telomere length. Furthermore, folic acid dose-dependently decreased Hcy, ROS, telomeric DNA oxidative damage, and apoptosis. 

Previous studies reported that folate deficiency is associated with apoptosis in neural stem cells and neurons [17,22]. The present study indicates that the same association occurs in astrocytes. Stimulation of apoptosis by folic acid deficiency may be due, in part, to the requirement of folate for nucleotide biosynthesis. However, this mechanism might not be a major cause of neurodegeneration in Alzheimer’s disease [14]. Instead, oxidative stress that is sensitive to folate levels and stimulates apoptosis may be a more important factor in neurodegeneration [23,24]. 

Folate deficiency inhibits the re-methylation process of Hcy metabolism and thereby causes Hcy to accumulate [11]. This mechanism may explain the present study’s finding that intracellular folate and Hcy concentrations varied inversely in astrocytes. The effect of folic acid on Hcy likely affects human health, because elevated Hcy levels evidently contribute to the pathogenesis of neurodegenerative disorders [12,13,14,15,16]. In particular, hyperhomocysteinemia is a risk factor for Alzheimer’s disease [15,16]. 

A previous study in primary astrocytes found that exposure to elevated Hcy concentrations caused endoplasmic reticulum stress and impaired autophagy that were reversed by supplementation of the cell culture medium with folate and vitamin B12 [25]. Another mechanism by which Hcy may stress astrocytes is by increasing intracellular ROS production. Excessive ROS levels cause oxidative damage in lipids, proteins, and especially in DNA, which may trigger apoptosis [26]. The present study found that intracellular Hcy and ROS levels decreased in parallel when folic acid treatment dose-dependently raised the intracellular folate concentration. Intracellular folate also alleviated apoptosis in animal and cell culture models of pathological oxidative stress that are not characterized by elevated Hcy concentrations [27]. 

Telomeres cap chromosome ends and are essential for genome stability, cell proliferation, and human health [28]. With each cell division, telomeres shorten until they reach a critical length, at which point a cell enters replicative senescence [29]. Shortened telomeres result in the exposure of the chromosomal ends, which are recognized as DNA double-strand breaks and activate the DNA damage response [30]. Critically short telomeres trigger cell senescence or apoptosis [31]. Oxidative stress accelerates telomere attrition [7,8,9,10]. Indeed, telomeres are preferential targets of oxidative stress due to their high content of guanine residues and the relative inefficiency of their DNA damage repair [8]. Oxidative stress has been proposed to accelerate telomere shortening and dysfunction through ROS-induced base damage in telomeric DNA [32]. However, there is controversy, since a recent study showed that acute 8-oxoG formation at telomeres did not induce telomere dysfunction [8]. The present study found in astrocytes that folic acid deficiency increased both 8-oxoG in telomeric DNA and telomere attrition, whereas high-dose folic acid decreased these parameters. 

In this study, 10 μmol/L of folic acid was used as the normal concentration. Dulbecco’s modified eagle medium (DMEM) has been recommended for the culture of primary astrocytes and it is also the traditional medium for the culture of neuronal cells. Since DMEM medium contains folic acid at 4 mg/mL (ca. 10 μmol/L) as a standard constituent, we chose 10 μmol/L folic acid as the normal concentration. However, the level used in this study is higher than the physiological level. Further study is needed to discuss the effect of folic acid at the physiological level in vivo.

This study has a limitation. This study hypothesized a mechanistic link between folate protection against astrocyte apoptosis and reduced telomere attrition and oxidative stress. The mechanism of oxidative stress-induced telomere attrition is only one of the potential mechanisms of folate protection. Further experiments are needed to demonstrate our hypothesis. However, astrocytes are the most widely distributed cells in the brain, and astrocyte apoptosis may contribute to the pathogenesis of neurodegenerative disorders such as Alzheimer’s disease. Also, folate metabolism is associated with the survival of astrocytes in the brain. The anti-apoptotic effect of folic acid on astrocytes may be exploited to improve neurodegenerative disorders prevention and therapeutic treatment.

## 4. Materials and Methods

### 4.1. Primary Astrocytes Culture

The experimental procedures were approved by the Tianjin Medical University Animal Ethics Committee (Authorization protocol number TMUaMEC 2019003). Primary astrocytes were prepared as described previously [33]. In brief, cerebral cortex was collected aseptically from 2-day-old SD rats. The pia mater was removed, and the remaining tissue was cut into 1 mm^3^ pieces and digested with 0.25% trypsin at 37 °C for 15 min. Cells were seeded in DMEM (Yuanpei, Shanghai, China) containing 10% fetal bovine serum (FBS) (Biological Industries, Beit Haemek, Israel), 10 μM folic acid (Sigma Aldrich, St. Louis, MO, USA), and 1000 mg/mL glucose. The cells were cultured in 95% air and 5% CO_2_ at 37 °C, and the culture medium was changed every 2 or 3 days. After 7 days, the oligodendrocytes and microglia were removed by thermostatic oscillation at 200 rpm for 18 h. Astrocytes were identified by immunofluorescence with an antibody against glial fibrillary acidic protein (GFAP) (Proteintech, Beijing, China). Cultures that were >95% GFAP-positive were used for the experiments (Figure 7). These primary astrocytes were incubated for 12 days with various concentrations of folic acid (0–40 μmol/L). Folic acid-free DMEM powder was purchased from Shanghai Yuanpei Biotechnology and combined with predetermined amounts of folic acid to prepare the culture media.

### 4.2. Cell Proliferation Assay

Cell proliferation was measured with the CellTiter 96^®^ AQueous One Solution Cell Proliferation assay (Promega Corporation, Madison, WI, USA). Cells were seeded at a density of 5000 cells/per well with 100 μL of culture medium in 96-well plates, then 20 μL CellTiter 96^®^ AQueous One Solution Reagent was pipetted into each well, and the plates were incubated for 4 h at 37 °C in a 5% CO_2_ incubator. Absorbance was recorded at 490 nm using a microplate reader (ELX800uv^™^; BioTek Instruments).

### 4.3. Cell Apoptosis Assay

Cell apoptosis was measured by flow cytometry and fluorescence staining with Hoechst 33342. Briefly, 1 × 10^6^ cells were collected, washed twice with D-PBS (PBS without calcium and magnesium), resuspended in 1×binding buffer, and incubated with 5 μL AnnexinV–FITC and 5 μL propidium iodide (PI) at room temperature for 20 min, followed by the addition of 400 μL of 1× binding buffer. The cells were sorted by a flow cytometer (BD FACSVerse, San Jose, CA, USA), and the data were analyzed with Flowjo 7.6. Nuclear morphology was detected by fluorescence staining using Hoechst 33342, and the data were analyzed with Image Pro Plus 6.0. Apoptotic cells were defined on the basis of their nuclear morphology, such as chromatin condensation and fragmentation. 

### 4.4. Intracellular Folate and Hcy Assays

Intracellular folate concentration was determined with the IMMULITE^®^ 2000 Folic Acid kit and an IMMULITE^®^2000 System Analyzer (Siemens, Berlin, Germany), according to the manufacturer’s instructions. The IMMULITE 2000 performs a 2-cycle, on-board sample treatment based on the principle of competitive immunoassay, which can detect folate, dihydrofolate, and tetrahydrofolate. Hcy concentration was measured with an ELISA kit (Cusabio Technology, Wuhan, China) in accordance with the manufacturer’s instructions. Microplates that had been precoated with antibodies specific for Hcy were read at a wavelength of 490 nm by a microplate reader (ELX800uv^™^; BioTek Instruments Inc, Winooski, VT, USA). The standard curve was generated using Curve Expert 1.3 software (Cusabio Technology, Wuhan, China). The intracellular concentrations of folate and Hcy were normalized to cell protein content, which was measured using a BCA total protein quantitative assay kit (Boster Biological Technology, Pleasanton, CA, USA).

### 4.5. Intracellular ROS Assay

Intracellular ROS levels were determined with 2′,7′-dichlorofluorescin diacetate staining and flow cytometry. Briefly, astrocytes were incubated with 10 μM 2′,7′-dichlorofluorescin diacetate (Sigma Aldrich, St. Louis, MO, USA) for 30 min in a humidified 5% CO_2_ atmosphere. The cells were washed thrice with PBS to remove extracellular fluorescence probes, then centrifuged and suspended in D-PBS. Fluorescence intensity was quantified with a flow cytometer (BD FACSVerse, San Jose, CA, USA) and FlowJo 7.6.

### 4.6. Measurement of 8-Oxoguanine (8-OxoG) in Telomeric DNA 

Genomic DNA was extracted using the Wizard^®^ Genomic DNA Purification kit (Promega Corporation, Madison, WI, USA) according to the manufacturer’s instructions. Diluted genomic DNA (approximately 20 ng/μL) was digested with formamidopyridine DNA glycosylase (FPG) (New England Biolabs, Arundel, Queensland, Australia) in the experimental samples, whereas water was substituted for FPG in the control samples. FPG cleaved oxidized guanine-containing DNA to create a fragmentary template for the subsequent PCR, and Ct values increased. All samples were set up on ice, then incubated at 37 °C for 16 h to allow exhaustive digestion. The 20 μL PCR mixture included 12.5 μL of SYBR Green qPCR Supermix (2×), 1 μL of DNA, 1 μL of telomere forward and reverse primers (10 µM), and PCR-grade water. The reaction mixtures were incubated at 95 °C for 10 min, followed by 40 cycles of 95 °C for 15 s and 60 °C for 1 min [34]. Primers specific for a telomeric gene (forward, 5′-CGGTTTGTTTGGGTTTGGGTTTGGGTTTGGGTTTGGGTT-3′; reverse, 5′-GGCTTGCCTTACCCTTACCCTTACCCTTACCCTTACCCT-3′) were used. The difference between the Ct values of FPG-treated versus untreated DNA represented the percentage of damaged DNA.

### 4.7. Telomeric DNA Oxidative Damage Assay 

Telomeric DNA oxidative damage was measured using immuno-fluorescence FISH, as described previously [35]. Astrocytes were grown on glass coverslips in 6-well plates, fixed with 4% paraformaldehyde for 20 min, exposed to 0.5% Triton X-100 for 20 min, and then hybridized with TelC–FITC (1:50, PNA Bio, Korea) at 37 °C for 12 h. The cells were washed twice with washing liquor (70% formamide; 1M Tris-HCl, pH = 7.2) and thrice with tris buffered saline with Tween (TBST). Blocking was done with 10% goat serum (BOSTER, China) for 1 h, followed by incubation with anti-phospho-histone H2AX (γ-H2AX) (Ser139) (1:500, Millipore, Massachusetts, USA) for 1 h and Cy3–conjugated Affinipure Goat Anti-Mouse IgG (H + L) (Proteintech, Beijing, China) (1:60) for 30 min. Images of cells were acquired on an inverted fluorescence microscope (Olympus, Tokyo, Japan). To determine whether 8-oxoG incorporation caused telomere dysfunction, γ-H2AX was used to mark telomere dysfunction-induced foci (TIFs). The cells were considered TIF-positive if they contained more than 2 immunofluorescent foci of both TelC–FITC and the DNA damage response protein γ-H2AX [36]. 

### 4.8. Measurement of Telomere Length 

Telomere length was determined by southern blot and quantitative real-time PCR (qPCR). Telomere restriction fragment (TRF) analysis was performed using a commercial kit (TeloTAGGG Telomere Length Assay, Roche Life Science, Mannheim, Switzerland), on the basis of the manufacturer’s instructions with slight modifications. Briefly, DNA (200 ng) was digested with MboI (New England Biolabs, Arundel, Queensland, Australia) 15 min at 37 °C, then electrophoresed on 1% agarose gel at 6 V for 14 h. The blotting membrane was washed, blocked, incubated with anti-DIG-alkaline phosphatase (1:4000 dilution, Roche, Mannheim, Switzerland) for 3 h, washed again, and exposed to a highly sensitive chemiluminescent substrate for 5 min (CDP-Star, Roche, Mannheim, Switzerland). After exposure of the blot to an X-ray film, terminal restriction fragment analysis was performed with Telo Tool version 1.3.

Additionally, telomere length was determined by qPCR, as proposed by Cawthon [37]. PCR was performed using the reference control gene (36B4 single-copy gene) and the telomeric gene. We used a PCR mixture of 25 μL including 12.5 μL of SYBR Green qPCR Supermix (2×), 1 μL of DNA, 1 μL of 36B4 forward and reverse primers (10 µM) or 1 μL of telomere forward and reverse primers (10 µM), and H_2_O (PCR-grade). The reaction mixtures were incubated at 95 °C for 5 min, then subjected to two cycles of 90 °C for 15 s, 49 °C for 15 s, which were followed by 40 amplification cycles (denaturation, 90 °C for 15 s; annealing, 62 °C for 10 s; extension, 74 °C for 15 s). Primers were specific for telomeric gene (forward, 5′-CGGTTTGTTTGGGTTTGGGTTTGGGTTTGGGTTTGGGTT-3′; reverse, 5′-GGCTTGCCTTACCCTTACCCTTACCCTTACCCTTACCCT-3′) and 36B4 (forward, 5′-ACTGGTCTAGGACCCGAGAAG-3′; reverse, 5′-TCAATGGTGCCTCTGGAGATT-3′). The assay was performed with a Roche 480 sequence detector. The expression of each gene was normalized to that of a reference control gene.

### 4.9. Statistical Analysis

Data were expressed as mean ± SEM values based on at least three independent experiments. The statistical software package SPSS 24.0 was used to evaluate differences between treatment groups by one-way ANOVA. When significant, ANOVA was followed by a post-hoc test (Tukey’s honestly significant difference test or Dunnet’s test). The correlation coefficients were calculated by spearman correlation analysis. A *p*-value less than 0.05 was considered statistically significant.

## 5. Conclusions

In conclusion, the novel results presented here are consistent with the hypothesis that folic acid decreases astrocyte apoptosis by preventing oxidative stress-induced telomere attrition. Since astrocyte apoptosis may contribute to the pathogenesis of Alzheimer’s disease and other neurodegenerative disorders [20,21], future studies should determine if the anti-apoptotic effect of folic acid on astrocytes can be exploited to improve disease prevention and therapeutic treatments.

## Figures and Tables

**Figure 1 ijms-21-00062-f001:**
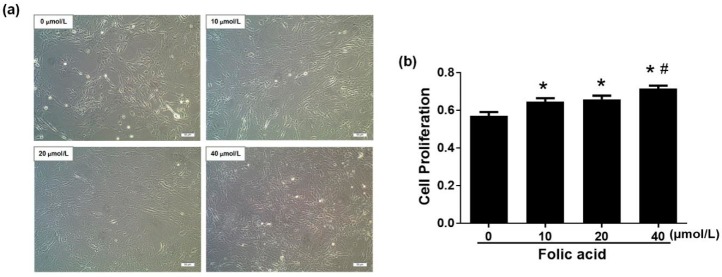
Folic acid increased cell proliferation in primary astrocytes. Primary cultures of rat astrocytes were incubated for 12 days with various concentrations of folic acid (0–40 μmol/L). (**a**) Cell morphology observed by light microscopy. Scale bar = 100 μm. (**b**) Bar graph of cell proliferation rates determined by the CellTiter 96^®^ AQueous One Solution Cell Proliferation Assay. The plotted values represent the mean ± SEM values of three experiments. * *p* < 0.05 compared with the folic acid-deficient group (0 μmol/L), ^#^
*p* < 0.05 compared with the normal-folic acid group (10 μmol/L).

**Figure 2 ijms-21-00062-f002:**
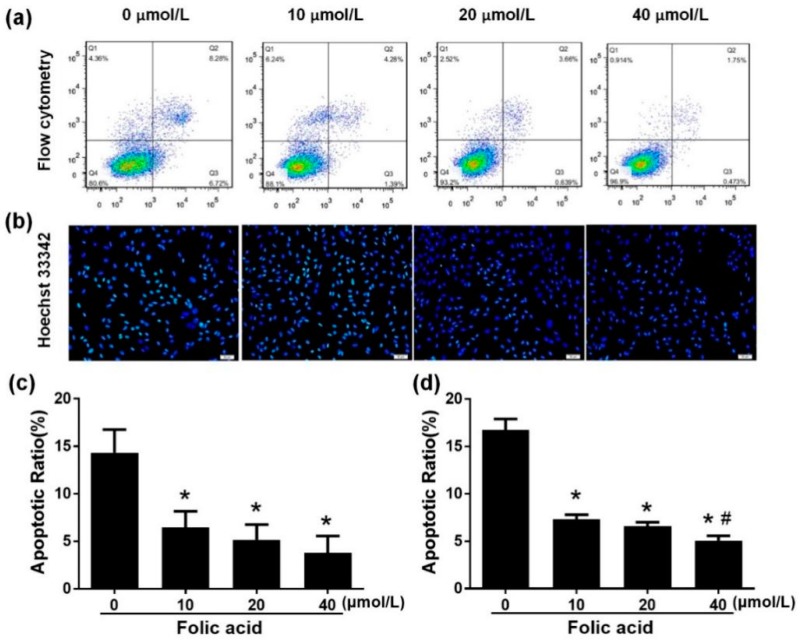
Folic acid decreased apoptosis in primary astrocytes. Primary astrocytes were incubated as described in Figure 1. (**a**) Scatter plots of apoptosis detected by flow cytometry (staining with AnnexinV–FITC and propidium iodide (PI)). Live cells appear in the bottom left square, early apoptotic cells appear in the bottom right square, and late apoptotic cells appear in the top right square. (**b**) Fluorescence staining of apoptotic cells by Hoechst 33342 (green) and nuclear staining by 4′,6-diamidino-2-phenylindole (DAPI) (blue). Scale bar = 100 μm. (**c**) Bar graph of total apoptotic rates determined by flow cytometry. (**d**) Bar graph of total apoptotic rates determined by Hoechst 33342 fluorescence staining. The plotted values represent the mean ± SEM values of three experiments. * *p* < 0.05 compared with the folic acid-deficient group (0 μmol/L), ^#^
*p* < 0.05 compared with the normal-folic acid group (10 μmol/L).

**Figure 3 ijms-21-00062-f003:**
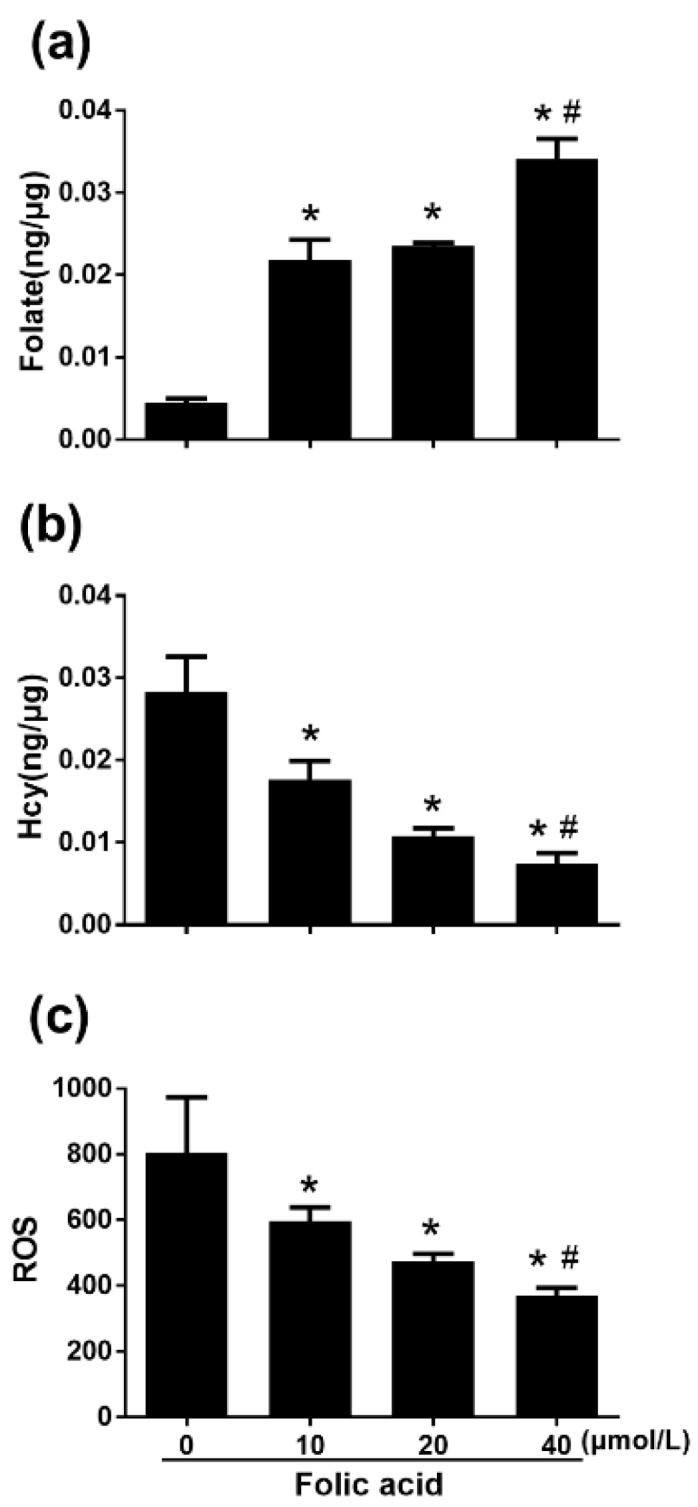
Folic acid increased intracellular folate concentration and decreased homocysteine (Hcy) and reactive oxygen species (ROS) levels. Primary astrocytes were incubated as described in Figure 1. Intracellular folate concentration was determined with the IMMULITE^®^ 2000 Folic Acid kit and an IMMULITE^®^2000 System analyzer. Hcy concentration was measured with an ELISA kit. Intracellular ROS levels were determined by 2′,7′-dichlorofluorescin diacetate staining and flow cytometry. (**a**) Bar graph of intracellular folate concentration. (**b**) Bar graph of intracellular Hcy concentration. (**c**) Bar graph of mean ROS intensity. The plotted values represent the mean ± SEM values of three separate experiments. * *p* < 0.05 compared with the folic acid-deficient group (0 μmol/L), ^#^
*p* < 0.05 compared with the normal-folic acid group (10 μmol/L).

**Figure 4 ijms-21-00062-f004:**
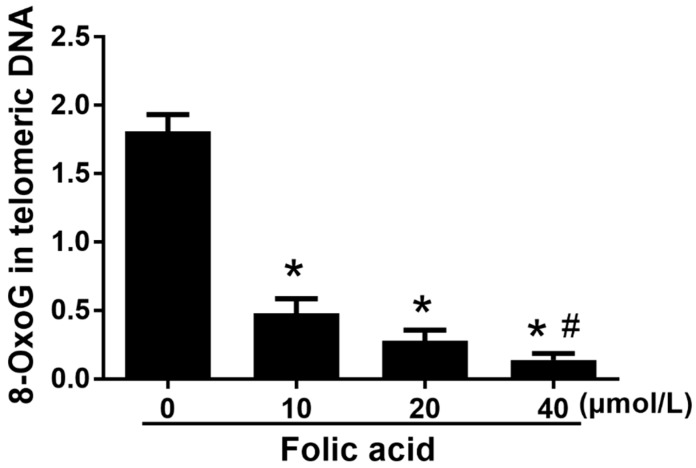
Folic acid treatment inhibited 8-oxoG incorporation in telomeric DNA. Primary astrocytes were incubated as described in Figure 1. RT-qPCR was used to detect 8-oxoG in telomeric DNA. The plotted values represent the mean ± SEM values of three separate experiments. * *p* < 0.05 compared with the folic acid-deficient group (0 μmol/L), ^#^
*p* < 0.05 compared with the normal-folic acid group (10 μmol/L).

**Figure 5 ijms-21-00062-f005:**
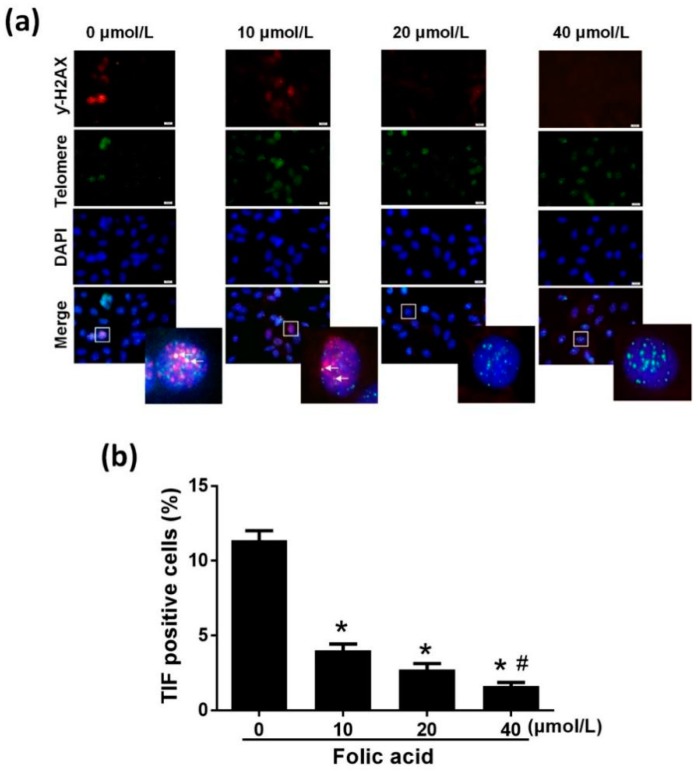
Folic acid decreased the abundance of telomere dysfunction-induced foci (TIF)-positive cells. Primary astrocytes were incubated as described in Figure 1. (**a**) Immunofluorescence FISH showing the co-localization of DNA double-strand breaks stained by γ-H2AX (red), telomeres stained by TelC–FITC (green), and nuclei stained by DAPI (blue). Scale bar = 10 μm. (**b**) Bar graph of the percentages of cells containing both TelC–FITC and γ-H2AX, i.e., TIF-positive cells, which showed by the arrow. More than 80 astrocytes per sample were scored. The plotted values represent the mean ± SEM values of three separate experiments. * *p* < 0.05 compared with the folic acid-deficient group (0 μmol/L), ^#^
*p* < 0.05 compared with the normal-folic acid group (10 μmol/L).

**Figure 6 ijms-21-00062-f006:**
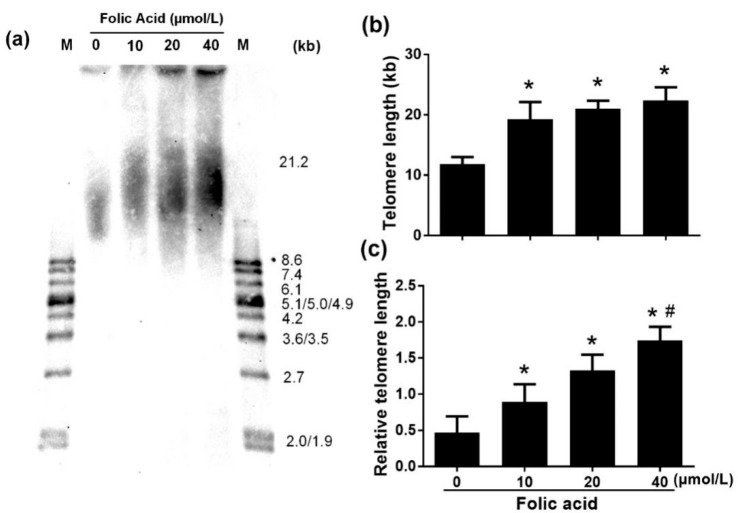
Folic acid inhibited telomere attrition. Primary astrocytes were incubated as described in Figure 1. (**a**) Mean telomere restriction fragments (TRF) detected by southern blot analysis. (**b**) Bar graph of southern blot densitometric analysis of mean TRF for genomic DNA. (**c**) Bar graph of relative telomere length determined by qPCR. The plotted values represent the mean ± SEM values of three separate experiments. * *p* < 0.05 compared with the folic acid-deficient group (0 μmol/L), ^#^
*p* < 0.05 compared with the normal-folic acid group (10 μmol/L).

**Figure 7 ijms-21-00062-f007:**
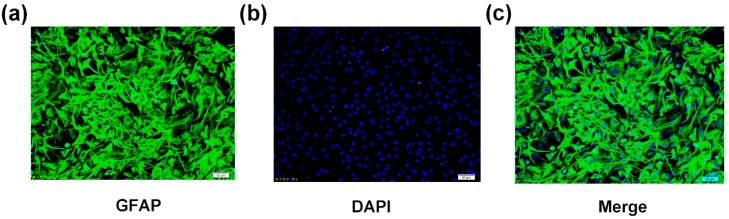
Identification of astrocytes. (**a**) Cells were stained with anti-glial fibrillary acidic protein (GFAP) antibody (green). Scale bar = 50 μm. (**b**) Cells were stained with DAPI (blue). Scale bar = 50 μm. (**c**) Merged.

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
