# Peer review of "Folic Acid Decreases Astrocyte Apoptosis by Preventing Oxidative Stress-Induced Telomere Attrition"

_ijms, 2019, doi:10.3390/ijms21010062_

Round 1

Reviewer 1 Report

This paper evaluated the hypothesis that folic acid decreases oxidative stress-induced astrocyte apoptosis by preventing telomere attrition. The findings showed that found that folic acid deficiency in primary astrocytes decreased intracellular folate, telomere length and cell proliferation, but it increased Hcy, ROS, telomeric DNA oxidative damage and apoptosis. Furthermore, folic acid dose-dependently decreased Hcy, ROS and telomeric DNA oxidative damage and apoptosis, and increased cell proliferation and telomere length

The study is well-written and clearly presented, the methodology is appropriate and the conclusions are believable .

I have only few comments:

-Did you perform any quality control for your assays? Please provide measures of reproducibility in order to provide information about the variation of the assay.

-Please describe more in details all statistical tests performed for the analysis of our data.

-Some discussion on the potential clinical implications of the findings would be also interesting.

Author Response

Comments from the Reviewer 1

This paper evaluated the hypothesis that folic acid decreases oxidative stress-induced astrocyte apoptosis by preventing telomere attrition. The findings showed that found that folic acid deficiency in primary astrocytes decreased intracellular folate, telomere length and cell proliferation, but it increased Hcy, ROS, telomeric DNA oxidative damage and apoptosis. Furthermore, folic acid dose-dependently decreased Hcy, ROS and telomeric DNA oxidative damage and apoptosis, and increased cell proliferation and telomere length

The study is well-written and clearly presented, the methodology is appropriate and the conclusions are believable.

I have only few comments:

Comment 1. Did you perform any quality control for your assays? Please provide measures of reproducibility in order to provide information about the variation of the assay.

Response: In this study, the data were expressed as mean ± SEM values based on at least three independent experiments. So, the data of this study can be reproducibility.

Comment 2. Please describe more in details all statistical tests performed for the analysis of our data.

Response: We rewrite the part of statistical analysis. And more details of statistic were added (P11, L303-303).

Added part of statistical analysis is below: “When significant, ANOVA was followed by a post hoc test (Tukey's honestly significant difference test or Dunnet's test). The correlation coefficients were calculated by spearman correlation analysis.”

Comment 3. Some discussion on the potential clinical implications of the findings would be also interesting.

Response: Astrocytes are the most widely distributed cells in the brain and astrocyte apoptosis may contribute to the pathogenesis of neurodegenerative disorders such as Alzheimer’s disease. And folate metabolism is associated with the survival of astrocytes in the brain. The anti-apoptotic effect of folic acid on astrocytes may be exploited to improve neurodegenerative disorders prevention and therapeutic treatment. We add those sentences in the part of Discussion (P8, L189-193).

Reviewer 2 Report

The manuscript shows that folic acid deficiency causes astrocyte apoptosis due to telomere attrition and this is linked to oxidative stress. The authors show that folic acid treatment rescue these effects.

The results of this study could be of interest, however major revisions and further experiments are required.

1 - The authors must demonstrate the purity of astrocyte cultures, showing GFAP-positive cells (as described in methods) or other specific markers, at least as supplemental figure.

2 - Figure 1 A is of very poor quality, I also suggest a greater magnification. It is very difficult to observe cell morphology and distinguish the described parameters (chromatin agglutination, karyopycnosis and nuclear fragmentation).

3 - Figure 1B: the increase of cell proliferation is minimal (1.1 times). How can it reach the statistical significance with only three experiments?

4 - Figure 1C and D: the legend of the figure and the text in the Results section are not clear. Please specify the used method also in the text and legend (described in Methods section), as the authors did for Hoechst staining.

5 - Why did the authors state that 10 micromole/L is the normal concentration? Please specify in the text the basis of this statement.

6 – Figure 2: Please specify used methods in the legend.

7 - Figure 5A is of poor quality. It is difficult to state that telomere length is higher in treated cells in this southern blot, particularly for the lanes 10 and 20 (please add the units). In figure 1B the length seems around 20Kb, but in the marker the 20 Kb is missing. Please, provide a better southern blot experiment as representative figure.

8 - The mechanistic link between folate protection against astrocyte apoptosis through reduced telomere attrition and oxidative stress may be only hypothesized by the authors in this study. Further experiments are needed to demonstrate the hypothesis. The discussion of the results then is limited. I suggest to better discuss the limitations of this study, particularly regarding the above mentioned link.

Author Response

Comments from the Reviewer 2

The manuscript shows that folic acid deficiency causes astrocyte apoptosis due to telomere attrition and this is linked to oxidative stress. The authors show that folic acid treatment rescue these effects.

The results of this study could be of interest, however major revisions and further experiments are required.

Comment 1. The authors must demonstrate the purity of astrocyte cultures, showing GFAP-positive cells (as described in methods) or other specific markers, at least as supplemental figure.

Response: Thank you for your comment. We added the figure of GFAP positive cells in the part of Materials and Methods as Figure 7 (P9, L210-212).

Comment 2. Figure 1 A is of very poor quality, I also suggest a greater magnification. It is very difficult to observe cell morphology and distinguish the described parameters (chromatin agglutination, karyopycnosis and nuclear fragmentation).

Response: We review the Figure 1a, and do the greater magnification for this figure. And we review the Figure 1a and b as the reviewed Figure 1, and Figure 1 c-f as the reviewed Figure 2 (P2-3).

Comment 3. Figure 1B: the increase of cell proliferation is minimal (1.1 times). How can it reach the statistical significance with only three experiments?

Response: We check this data the F value is 12.860, and p value is 0.002 for the difference of treatment groups in cell proliferation assay by ANOVA. Then followed by a post hoc test, p=0.021 for the high dose folic acid group (40 μmol/L) compare with the normal folic acid group (10 μmol/L), it reached the statistical significance even if only 1.1 times change.

Comment 4. Figure 1C and D: the legend of the figure and the text in the Results section are not clear. Please specify the used method also in the text and legend (described in Methods section), as the authors did for Hoechst staining.

Response: We add the specify the used method in the text and legend of reviewed Figure 2 (P3, L83-90).

Comment 5. Why did the authors state that 10 micromole/L is the normal concentration? Please specify in the text the basis of this statement.

Response: DMEM medium has been recommended for the culture of primary astrocytes, and DMEM is also tradition medium for the culture of neuronal cells. Due to DMEM medium contain folic acid at 4 mg/mL (ca. 10 μmol/L) as a standard constituent, we chose 10 μmol/L folic acid as the normal concentration. However, the level used in this study is higher than that physiology level. Further study is needed to discuss the effect of folic acid on physiology level in vivo.

Comment 6. Figure 2: Please specify used methods in the legend.

Response: We have added the specify of methods which detect the intracellular concentrations of folate, Hcy and ROS (P4, L102-105).

Added part is below: “Intracellular folate concentration was determined with the IMMULITE® 2000 Folic Acid kit and an IMMULITE®2000 System Analyzer. Hcy concentration was measured with an ELISA kit. Intracellular ROS levels were determined with 2’,7’-dichlorofluorescin diacetate staining and flow cytometry.”

Comment 7. Figure 5A is of poor quality. It is difficult to state that telomere length is higher in treated cells in this southern blot, particularly for the lanes 10 and 20 (please add the units). In figure 1B the length seems around 20 Kb, but in the marker the 20 Kb is missing. Please, provide a better southern blot experiment as representative figure.

Response: We reviewed the Figure 5a as the reviewed Figure 6a, increased the figure quality and added the marker (P7, L140).

Comment 8. The mechanistic link between folate protection against astrocyte apoptosis through reduced telomere attrition and oxidative stress may be only hypothesized by the authors in this study. Further experiments are needed to demonstrate the hypothesis. The discussion of the results then is limited. I suggest to better discuss the limitations of this study, particularly regarding the above-mentioned link.

Response: Thank you for your comments. We agree with the reviewer. The mechanistic link between folate protection against astrocyte apoptosis through reduced telomere attrition and oxidative stress may be only hypothesize, and only one of potential mechanisms. So, we added some sentences in the discussion part to discuss this limitation (P8, L186-189).

Added part is below: “There also a limitation of this study. This study hypothesized mechanistic link between folate protection against astrocyte apoptosis through reduced telomere attrition and oxidative stress. The mechanism of oxidative stress-induced telomere attrition is only one of potential mechanisms. Further experiments are needed to demonstrate the hypothesis.”

Round 2

Reviewer 2 Report

Comment 1. The authors must demonstrate the purity of astrocyte cultures, showing GFAP-positive cells (as described in methods) or other specific markers, at least as supplemental figure.

Response: Thank you for your comment. We added the figure of GFAP positive cells in the part of Materials and Methods as Figure 7 (P9, L210-212)

Reply to response 1: Ok

Comment 2. Figure 1 A is of very poor quality, I also suggest a greater magnification. It is very difficult to observe cell morphology and distinguish the described parameters (chromatin agglutination, karyopycnosis and nuclear fragmentation).

Response: We review the Figure 1a, and do the greater magnification for this figure. And we review the Figure 1a and b as the reviewed Figure 1, and Figure 1 c-f as the reviewed Figure 2 (P2-3).

Reply to response 2: I am still perplexed about the significance of Fig 1A, with regard to the fact that folic acid deficiency increases chromatic agglutination, karyopyknosis and nuclear fragmentation compared with high folic acid groups. The cell morphology with this magnification does not support what the authors claim, the figure only documents the decrease of cell proliferation. Please modify the text of results.

Comment 3. Figure 1B: the increase of cell proliferation is minimal (1.1 times). How can it reach the statistical significance with only three experiments?

Response: We check this data the F value is 12.860, and p value is 0.002 for the difference of treatment groups in cell proliferation assay by ANOVA. Then followed by a post hoc test, p=0.021 for the high dose folic acid group (40 μmol/L) compare with the normal folic acid group (10 μmol/L), it reached the statistical significance even if only 1.1 times change.

Reply to response 3: Ok

Comment 4. Figure 1C and D: the legend of the figure and the text in the Results section are not clear. Please specify the used method also in the text and legend (described in Methods section), as the authors did for Hoechst staining.

Response: We add the specify the used method in the text and legend of reviewed Figure 2 (P3, L83-90).

Reply to response 4: Ok

Comment 5. Why did the authors state that 10 micromole/L is the normal concentration? Please specify in the text the basis of this statement.

Response: DMEM medium has been recommended for the culture of primary astrocytes, and DMEM is also tradition medium for the culture of neuronal cells. Due to DMEM medium contain folic acid at 4 mg/mL (ca. 10 μmol/L) as a standard constituent, we chose 10 μmol/L folic acid as the normal concentration. However, the level used in this study is higher than that physiology level. Further study is needed to discuss the effect of folic acid on physiology level in vivo.

Reply to response 5: Ok, however it is better to specify and discuss this concept.

Comment 6. Figure 2: Please specify used methods in the legend.

Response: We have added the specify of methods which detect the intracellular concentrations of folate, Hcy and ROS (P4, L102-105).

Added part is below: “Intracellular folate concentration was determined with the IMMULITE® 2000 Folic Acid kit and an IMMULITE®2000 System Analyzer. Hcy concentration was measured with an ELISA kit. Intracellular ROS levels were determined with 2’,7’-dichlorofluorescin diacetate staining and flow cytometry.”

Reply to response 6: ok

Comment 7. Figure 5A is of poor quality. It is difficult to state that telomere length is higher in treated cells in this southern blot, particularly for the lanes 10 and 20 (please add the units). In figure 1B the length seems around 20 Kb, but in the marker the 20 Kb is missing. Please, provide a better southern blot experiment as representative figure.

Response: We reviewed the Figure 5a as the reviewed Figure 6a, increased the figure quality and added the marker (P7, L140).

Reply to response 7: ok 

Comment 8. The mechanistic link between folate protection against astrocyte apoptosis through reduced telomere attrition and oxidative stress may be only hypothesized by the authors in this study. Further experiments are needed to demonstrate the hypothesis. The discussion of the results then is limited. I suggest to better discuss the limitations of this study, particularly regarding the above-mentioned link.

Response: Thank you for your comments. We agree with the reviewer. The mechanistic link between folate protection against astrocyte apoptosis through reduced telomere attrition and oxidative stress may be only hypothesize, and only one of potential mechanisms. So, we added some sentences in the discussion part to discuss this limitation (P8, L186-189).

Added part is below: “There also a limitation of this study. This study hypothesized mechanistic link between folate protection against astrocyte apoptosis through reduced telomere attrition and oxidative stress. The mechanism of oxidative stress-induced telomere attrition is only one of potential mechanisms. Further experiments are needed to demonstrate the hypothesis.”

Reply to response 8: ok 

Author Response

Reply to response 1: ok

Comment 2. Figure 1 A is of very poor quality, I also suggest a greater magnification. It is very difficult to observe cell morphology and distinguish the described parameters (chromatin agglutination, karyopycnosis and nuclear fragmentation).

Response: We review the Figure 1a, and do the greater magnification for this figure. And we review the Figure 1a and b as the reviewed Figure 1, and Figure 1 c-f as the reviewed Figure 2 (P2-3).

Reply to response 2: I am still perplexed about the significance of Fig 1A, with regard to the fact that folic acid deficiency increases chromatic agglutination, karyopyknosis and nuclear fragmentation compared with high folic acid groups. The cell morphology with this magnification does not support what the authors claim, the figure only documents the decrease of cell proliferation. Please modify the text of results.

Response: Thank you for your comments. We have changed the text of results of Fig 1a (P2, L64-65).

The reviewed part is showed below: “Folic acid deficiency decreased cell proliferation compared with high folic acid groups (20 or 40 μmol/L) (Figure 1a).”

Reply to response 3: ok

Reply to response 4: ok

Comment 5. Why did the authors state that 10 micromole/L is the normal concentration? Please specify in the text the basis of this statement.

Response: DMEM medium has been recommended for the culture of primary astrocytes, and DMEM is also tradition medium for the culture of neuronal cells. Due to DMEM medium contain folic acid at 4 mg/mL (ca. 10 μmol/L) as a standard constituent, we chose 10 μmol/L folic acid as the normal concentration. However, the level used in this study is higher than that physiology level. Further study is needed to discuss the effect of folic acid on physiology level in vivo.

Reply to response 5: Ok, however it is better to specify and discuss this concept.

Response: We add some sentences in the part of discussion (P8, L186-191).

Added part is below: “In this study, 10 μmol/L folic acid used as the normal concentration. DMEM medium has been recommended for the culture of primary astrocytes, and DMEM is also tradition medium for the culture of neuronal cells. Due to DMEM medium contain folic acid at 4 mg/mL (ca. 10 μmol/L) as a standard constituent, we chose 10 μmol/L folic acid as the normal concentration. However, the level used in this study is higher than that physiology level. Further study is needed to discuss the effect of folic acid on physiology level in vivo.”

Reply to response 6: ok

Reply to response 7: ok

Reply to response 8: ok